

# Sarcopenia is not associated with inspiratory muscle strength but with expiratory muscle strength among older adults requiring long-term care/support

Yohei Sawaya[1,2,*], Takahiro Shiba[2,*], Masahiro Ishizaka[1], Tamaki Hirose[1,2], Ryo Sato[2], Akira Kubo[1] and Tomohiko Urano[2,3]

[1] Department of Physical Therapy, School of Health Sciences, International University of Health and Welfare, Otawara, Tochigi, Japan
[2] Nishinasuno General Home Care Center, Department of Day Rehabilitation, Care Facility for the Elderly "Maronie-en", Nasushiobara, Tochigi, Japan
[3] Department of Geriatric Medicine, School of Medicine, International University of Health and Welfare, Narita, Chiba, Japan
* These authors contributed equally to this work.

## ABSTRACT

**Background:** Recently, the concept of respiratory sarcopenia has been advocated, but evidence is lacking regarding which respiratory parameters are appropriate indicators. Therefore, we investigated the association between sarcopenia, respiratory function, and respiratory muscle strength to identify the most appropriate parameters for respiratory sarcopenia.

**Methods:** We included 124 older adults (67 men, 57 women; average age 77.2 ± 8.3 years) requiring long-term care/support who underwent Day Care for rehabilitation. Handgrip strength, usual gait speed, and skeletal muscle mass were measured using bioelectrical impedance analysis. Participants were then diagnosed with sarcopenia using the algorithm of the Asian Working Group for Sarcopenia 2019. Parameters of respiratory function (forced vital capacity, forced expiratory volume in one second [FEV1.0], FEV1.0%, and peak expiratory flow rate) and respiratory muscle strength (maximal expiratory pressure [MEP] and maximal inspiratory pressure) were also measured according to American Thoracic Society guidelines. Respiratory parameters significantly related to sarcopenia were identified using binomial logistic regression and receiver operating characteristic analyses.

**Results:** Seventy-seven participants were classified as having sarcopenia. Binomial logistic regression analysis showed that MEP was the only respiratory parameter significantly associated with sarcopenia. The cut-off MEP value for predicting sarcopenia was 47.0 $cmH_2O$ for men and 40.9 $cmH_2O$ for women.

**Conclusions:** The most appropriate parameter for assessing respiratory sarcopenia may be MEP, which is an indicator of expiratory muscle strength, rather than FVC, MIP, or PEFR, as suggested in previous studies. Measuring MEP is simpler than measuring respiratory function parameters. Moreover, it is expected to have clinical applications such as respiratory sarcopenia screening.

Corresponding authors
Yohei Sawaya, sawaya@iuhw.ac.jp
Tomohiko Urano,
turano@iuhw.ac.jp

# INTRODUCTION

Respiratory sarcopenia has been recently defined as "whole-body sarcopenia, and low respiratory muscle mass followed by low respiratory muscle strength and/or deteriorated respiratory function" (*Nagano et al., 2021*). However, it is difficult to assess low respiratory muscle mass as no cut-off value for it has been established (*Nagano et al., 2021*). Therefore, criteria other than respiratory muscle mass are used to diagnose respiratory sarcopenia (*Nagano et al., 2021*). The indicators used to assess respiratory sarcopenia were maximal inspiratory pressure (MIP) and forced vital capacity (FVC). However, *Kera et al. (2019)* who were the first to define respiratory sarcopenia, used peak expiratory flow rate (PEFR) as a respiratory parameter for evaluating respiratory sarcopenia. Several other respiratory parameters have been reported to be associated with sarcopenia, but it is unclear which one is the most appropriate for respiratory sarcopenia (*Ohara et al., 2018*; *Ohara et al., 2020*; *Ridwan et al., 2021*; *Mishra et al., 2020*). The review also clearly stated that "there is a lack of evidence in the specificity of the measures and its cut-off values" (*Nagano et al., 2021*). Thus, further accumulation of evidence is essential.

Respiratory parameters are divided into those used to assess respiratory function, such as FVC and PEFR, or respiratory muscle strength. Respiratory muscle strength is considered to be the total muscle strength of the diaphragm and other respiratory muscles and is defined as the maximal expiratory pressure (MEP) and MIP (*Cook, Mead & Orzalesi, 1964*). However, no study has simultaneously analyzed the association between sarcopenia, respiratory function, and respiratory muscle strength.

Regarding the respiratory muscles, the main inspiratory muscle is the diaphragm (*De Troyer & Estenne, 1988*; *Epstein, 1994*). The diaphragm differs from skeletal muscles considering its special embryological characteristics and its ability to continuously contract and relax (*Jinguji, 1978*; *Fujishima et al., 2019*). The muscles of deglutition, which show similar characteristics, are thought to be less prone to muscle atrophy than skeletal muscles, and this difference should be considered regarding sarcopenia for these muscles (*Fujishima et al., 2019*). In addition, the diaphragm has a high percentage of fatigue-resistant fibers (*Epstein, 1994*). Based on the above, we hypothesized that the diaphragm might be less likely to develop respiratory sarcopenia. Furthermore, because our prior fundamental study showed that MEP was independently associated with skeletal muscle mass index (SMI), we also hypothesized that sarcopenia is more associated with expiratory muscle strength than inspiratory muscle strength (*Sawaya et al., 2020*).

Therefore, this study aimed to determine the most appropriate parameter for assessing respiratory sarcopenia by measuring both respiratory function and respiratory muscle strength. Moreover, we aimed to calculate the cut-off value for the obtained parameter.

## MATERIALS AND METHODS

### Study design

This was a cross-sectional study conducted at a single daycare between March 2018 and August 2019. All participants were informed of the study orally and in writing, and written consent was obtained from all participants. The Ethical Review Committee of the International University of Health and Welfare approved this study (Approval No.: 17-Io-189-7). The study complied with the principles of the Declaration of Helsinki.

### Study participants

This study included 154 community-dwelling older adults aged 60 years or older who were under Day Care. All participants had been certified as requiring long-term care/support among the Japanese system (*Yamada & Arai, 2020*). Our Day Care for older adults involves rehabilitation services that include exercise, transportation, meals, and bathing options. The study included those whose body composition parameters could be measured in the standing position. We excluded those who had been diagnosed as having dementia or aphasia, those for whom performing spirometry was difficult, and those with respiratory diseases such as those on home oxygen therapy or with a forced expiratory volume % in 1 s (FEV1.0%) of <70%, considering the effect of airway obstruction on respiratory function according to a previous study (*Kera et al., 2019*).

### Sarcopenia assessment

Sarcopenia was diagnosed based on the algorithm of the Asian Working Group for Sarcopenia (AWGS) 2019 (*Chen et al., 2020*), which was characterized by low skeletal muscle mass, low muscle strength, and/or low physical function. Grip strength was measured twice (Smedley-type hand dynamometer, TKK 5401 Grip-D, Takei Scientific Instruments, Japan) on each side in the sitting position, and the maximum value was used as the representative value. Usual gait speed was measured once at a distance of 5 m using a stopwatch (*Arai, 2018*). In addition to the measurement section of 5 m, acceleration and deceleration paths were set as walking paths. Skeletal and trunk muscle mass were measured using a multifrequency bioelectrical impedance analysis (BIA) body composition analyzer (InBody 520; InBody, Japan), and the skeletal muscle mass index (SMI) was calculated by dividing the skeletal muscle mass of limbs by the square of the height. The cutoff for each measurement is as follows: grip strength <28.0 kg for men and <18.0 kg for women, usual gait speed <1.0 m/s for both men and for women, and SMI <7.0 kg/m$^2$ for men and <5.7 kg/m$^2$ for women (*Chen et al., 2020*).

### Respiratory function and muscle strength

Respiratory function and muscle strength were measured using a spirometer (Autospiro AS-507; Minato, Japan) and an attached unit (AAM377, Minato, Japan). All measurements were performed by physical therapists based on the American Thoracic Society (ATS)/European Respiratory Society (ERS) guidelines (*American Thoracic Society/ European Respiratory Society, 2002*). Respiratory function mode involved FVC, forced expiratory volume in one second (FEV1.0), FEV1.0%, and PEFR. The respiratory muscle

strength mode involved MEP and MIP. Respiratory function was measured first, followed by respiratory muscle strength. The maximum value after three measurements was used as the representative value.

## Basic attributes

Details regarding age, height, care level, smoking history, and morbidity were collected from the medical records of the Day Care. Since long-term certification is classified into seven care levels, this study used an ordinal scale, with a ranking from 1 (mild) to 7 (severe) (*Yamada & Arai, 2020*). Body weight was obtained from the data of body composition measurement.

## Statistical analyses

First, the basic attributes and measurement values of the sarcopenia and non-sarcopenia groups were compared using the unpaired t-test, Wilcoxon rank-sum test, chi-square test, and Fisher's exact test. Second, binomial logistic regression analysis using stepwise selection was conducted with the presence or absence of sarcopenia as the dependent variable. Respiratory parameters, after considering multicollinearity, were considered as independent variables. Sex, age, care level, body mass index, and morbidity were used as control variables. Third, for the respiratory parameters extracted by binomial logistic regression analysis, the area under the curve (AUC), sensitivity, and specificity were calculated using the Youden Index method from the receiver operating characteristic (ROC) curve, and the cut-off value of sarcopenia was determined. Fourth, the MEP and MIP components were analyzed using partial correlation, controlling for sex and age. The partial correlation was performed after confirming the normal distribution of trunk muscle mass per height squared and SMI by Kolmogorov–Smirnov test and Shapiro–Wilk test. Statistical analysis was performed using SPSS version 25 (IBM Japan, Japan), with a significance level of 5%. Power analysis was performed using G*Power version 3.1.9.2 (*Faul et al., 2007*).

## RESULTS

Figure 1 shows the flowchart of the study participants. After applying the exclusion criteria, there were 124 eligible participants, 77 with sarcopenia and 47 with non-sarcopenia. Table 1 shows a comparison of the basic attributes and measurement values between the sarcopenia and non-sarcopenia groups. The sarcopenia group had a significantly lower MEP for men and PEFR and MEP for women. There was no significant difference in morbidity between the sarcopenia and non-sarcopenia groups.

Binomial logistic regression analysis showed that MEP was the only respiratory parameter that was significantly associated with sarcopenia (Table 2). Four parameters (FVC, FEV1.0%, PEFR, and MIP) were excluded from the independent variables by the stepwise selection method. FEV1.0 was excluded after multicollinearity was considered. Post-hoc power analysis was performed for binomial logistic regression analysis. For the multiple linear regression of F tests, the power was 1.00, based on the effect size of 0.39, calculated from the regression equation (Nagelkerke $R^2$ = 0.281). Analyses by sex are

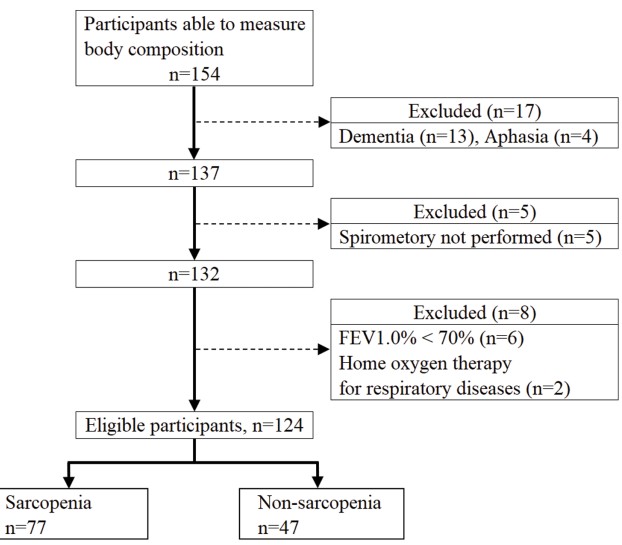

**Figure 1  Flowchart of participant recruitment.**     

shown in Tables S1 and S2. Figure 2 shows ROC curves. The cut-off value of MEP for predicting sarcopenia was 47.0 cmH$_2$0 in men (AUC 0.689, sensitivity 91.7%, specificity 41.9%, $P$ = 0.011) and 40.9 cmH$_2$0 in women (AUC 0.677, sensitivity 65.2%, specificity 70.6%, $P$ = 0.024).

Table 3 shows the results of the partial correlation analysis. MEP and MIP showed a significant positive correlation (r = 0.550) with each other. MEP also showed a significant positive correlation with grip strength, usual gait speed, and SMI, while MIP was not significantly associated with these parameters.

## DISCUSSION

In previous reports on sarcopenia and respiratory parameters, skeletal muscle mass was calculated from predictive equations (*Ohara et al., 2018*; *Ohara et al., 2020*; *Ridwan et al., 2021*), and original cut-off values were used for grip strength, walking speed, and SMI for sarcopenia (*Kera et al., 2019*). This study is the first report to combine (1) compliance with the AWGS2019 criteria, (2) skeletal muscle mass measurement using the BIA method recommended by AWGS and EWGSOP (*Chen et al., 2020*; *Cruz-Jentoft et al., 2019*), (3) simultaneous analysis of respiratory function and respiratory muscle strength, and (4) compliance with ATS guidelines for spirometry instruments (*American Thoracic Society/European Respiratory Society, 2002*). Our results showed the lack of significant association between sarcopenia and MIP and FVC as indicators of respiratory sarcopenia and a significant association between sarcopenia and MEP. These findings may be explained by muscle atrophy of the diaphragm and rectus abdominis as well as the component factors of MEP.

Sarcopenia was not associated with inspiratory muscle strength of inspiration but with expiratory muscle strength. Muscle fibers are classified into type 1 fibers (slow-oxidative and fatigue-resistant), type 2A fibers (fast-oxidative-glycolytic and fatigue-resistant), and type 2B fibers (fast-glycolytic and fatigue-sensitive) (*Epstein, 1994*).

**Table 1 Basic attributes and measurements values with and without sarcopenia.**

| | Men (n = 67) | | | Women (n = 57) | | |
|---|---|---|---|---|---|---|
| | Sarcopenia (n = 43) | Non-sarcopenia (n = 24) | P-value | Sarcopenia (n = 34) | Non-sarcopenia (n = 23) | P-value |
| Age (years) | 77.5 ± 7.9 | 73.4 ± 7.6 | 0.041* | 77.4 ± 8.0 | 80.0 ± 9.5 | 0.269 |
| Height (cm) | 161.3 ± 5.4 | 165.6 ± 6.2 | 0.004* | 150.3 ± 5.7 | 153.9 ± 5.2 | 0.019* |
| Weight (kg) | 57.6 ± 6.6 | 65.6 ± 7.8 | <0.001* | 46.3 ± 8.0 | 56.6 ± 7.9 | <0.001* |
| BMI (kg/m$^2$) | 22.2 ± 2.6 | 24.0 ± 3.0 | 0.012* | 20.6 ± 4.0 | 23.9 ± 3.1 | 0.001* |
| Care level (1–7)[†] | 3 (2–3) | 3 (2–4) | 0.659 | 3 (2–4) | 2 (1–3) | 0.011** |
| Smoking history[††] | 28 | 12 | 0.411 | 1 | 0 | 1.000 |
| Sarcopenia assessment | | | | | | |
| Grip strength (kg) | 22.9 ± 5.6 | 31.0 ± 5.0 | <0.001* | 15.6 ± 4.8 | 19.0 ± 4.6 | 0.010* |
| Usual gait (m/s) | 0.64 ± 0.27 | 0.77 ± 0.34 | 0.085 | 0.52 ± 0.24 | 0.68 ± 0.39 | 0.084 |
| SMI (kg/m$^2$) | 6.07 ± 0.57 | 7.40 ± 0.78 | <0.001* | 5.13 ± 0.44 | 6.28 ± 0.48 | <0.001* |
| Trunk muscle mass per square height (kg/m$^2$) | 6.99 ± 0.75 | 7.84 ± 0.90 | <0.001* | 6.29 ± 0.57 | 6.88 ± 0.62 | 0.001* |
| Respiratory function | | | | | | |
| FVC (L) | 2.23 ± 0.68 | 2.52 ± 0.64 | 0.096 | 1.55 ± 0.40 | 1.66 ± 0.43 | 0.352 |
| FEV1.0 (L) | 1.88 ± 0.56 | 2.10 ± 0.59 | 0.120 | 1.35 ± 0.35 | 1.41 ± 0.37 | 0.561 |
| FEV1.0% (%) | 84.5 ± 7.4 | 83.8 ± 8.5 | 0.736 | 87.2 ± 6.9 | 84.8 ± 7.1 | 0.197 |
| PEFR (L/s) | 4.20 ± 1.48 | 4.69 ± 1.90 | 0.244 | 2.91 ± 1.03 | 3.56 ± 1.26 | 0.038* |
| Respiratory muscle strength | | | | | | |
| MEP (cmH$_2$O) | 52.8 ± 18.4 | 67.7 ± 23.9 | 0.006* | 36.2 ± 12.6 | 45.7 ± 14.6 | 0.011* |
| MIP (cmH$_2$O) | 38.3 ± 20.6 | 43.6 ± 16.2 | 0.282 | 29.8 ± 10.8 | 33.0 ± 18.2 | 0.464 |
| Morbidity | | | | | | |
| Hypertension | 17 | 10 | 1.000 | 15 | 10 | 1.000 |
| Cerebrovascular dis | 23 | 16 | 0.317 | 14 | 7 | 0.576 |
| Orthopedic dis | 22 | 9 | 0.317 | 25 | 17 | 1.000 |
| Cancer | 11 | 4 | 0.545 | 5 | 4 | 1.000 |
| Intractable neurological dis | 9 | 4 | 0.757 | 5 | 1 | 0.385 |

Notes:
* $P < 0.05$ for unpaired t test.
** $P < 0.05$ for Wilcoxon rank-sum test.
[†] Median (25th percentile–75th percentile).
[††] With missing data (63 men, 55 women).
BMI, body mass index; SMI, skeletal muscle mass index; FVC, forced vital capacity; FEV1.0, forced expiratory volume in 1 s; PEFR, peak expiratory flow rate; MEP, maximal expiratory pressure; MIP, maximal inspiratory pressure; dis, disease.

The diaphragm, which is the main inspiratory muscle, is composed of 80% fatigue-resistant fibers (55% type 1, 25% type 2A), while the rectus abdominis, the main expiratory muscle, is composed of 46% type 2B fibers (*Epstein, 1994*; *Ito, 1998*). In sarcopenia, muscle atrophy is selectively observed in fast-twitch fibers (*Lexell, Taylor & Sjöström, 1988*), suggesting that the diaphragm may be less atrophic and maintain its inspiratory muscle strength, while the rectus abdominis may be more prone to atrophy and loss of expiratory muscle strength. These supportive reports indicate that the diaphragm shows very little change in muscle mass and strength with aging (*Krumpe et al., 1985*; *Caskey et al., 1989*; *Polkey et al., 1997*; *Mizuno, 1991*). Although a study reported that the

**Table 2 Association between sarcopenia and respiratory parameters based on binomial logistic regression analysis.**

|  | β | P-value | Odds ratio | 95% CI |
|---|---|---|---|---|
| MEP (cmH$_2$O) | −0.028 | 0.034 | 0.973 | [0.948–0.998] |
| Sex | −0.988 | 0.056 | 0.373 | [0.135–1.028] |
| Age (years) | −0.004 | 0.868 | 0.996 | [0.945–1.049] |
| BMI (kg/m$^2$) | −0.239 | 0.002 | 0.788 | [0.677–0.916] |
| Certification (1–7) | 0.323 | 0.052 | 1.381 | [0.998–1.911] |
| Hypertension | 0.275 | 0.540 | 1.316 | [0.547–3.171] |
| Cerebrovascular disease | −0.116 | 0.819 | 0.890 | [0.330–2.400] |
| Orthopedic disease | 0.248 | 0.608 | 1.282 | [0.496–3.310] |
| Cancer | −0.243 | 0.655 | 0.784 | [0.270–2.278] |
| Intractable neurological disease | 0.037 | 0.959 | 1.037 | [0.258–4.167] |

**Notes:**
CI, confidence interval; MEP, maximal expiratory pressure; BMI, body mass index.
Dependent variables: Non-sarcopenia = 0, sarcopenia = 1.
Independent variables: Men = 0, women = 1; without morbidity = 0, with morbidity = 1.
Nagelkerke R$^2$ = 0.281.

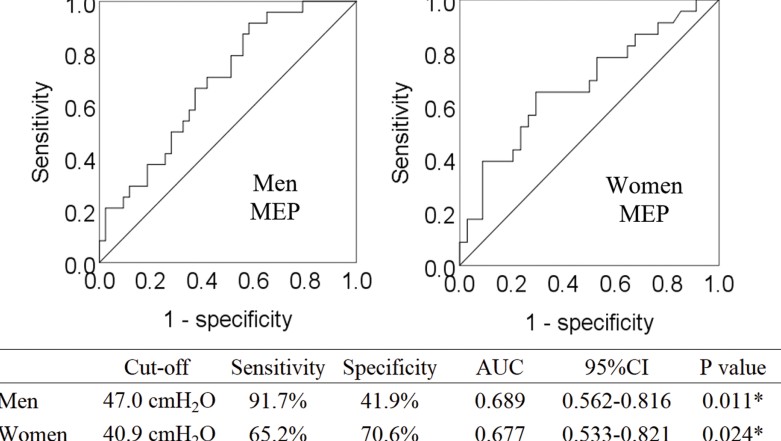

|  | Cut-off | Sensitivity | Specificity | AUC | 95%CI | P value |
|---|---|---|---|---|---|---|
| Men | 47.0 cmH$_2$O | 91.7% | 41.9% | 0.689 | 0.562-0.816 | 0.011* |
| Women | 40.9 cmH$_2$O | 65.2% | 70.6% | 0.677 | 0.533-0.821 | 0.024* |

MEP, maximal expiratory pressure; AUC, area under the curve; CI, confidence interval.
* P < 0.05.

**Figure 2 Receiver operating characteristic curve analysis for identifying sarcopenia *via* MEP.**

diaphragm is significantly thinner in older adults with sarcopenia, it is not clear whether the change is clinically significant because only a difference of 0.2–0.4 mm was noted (*Deniz et al., 2021*).

Next, we performed a partial correlation analysis, focusing on the component factors of MEP and MIP. Although the MEP and MIP of the participants in this study were correlated, only MEP showed a significant association with grip strength, walking speed, and SMI, which are indicators of sarcopenia. Trunk muscle mass was also found to have a significant association only with MEP. These results suggest that the component factors of MEP and MIP are different, and that MEP may contain sarcopenia elements,

**Table 3 Partial correlation of MEP and MIP with component factors.**

|  | MEP | P-value | MIP | P-value |
|---|---|---|---|---|
| Height (cm) | 0.116 | 0.203 | 0.019 | 0.834 |
| Weight (kg) | 0.370 | <0.001 | 0.204 | 0.024 |
| BMI (kg/m$^2$) | 0.315 | <0.001 | 0.199 | 0.028 |
| Grip strength (kg) | 0.263 | 0.003 | 0.075 | 0.411 |
| Usual gait (m/s) | 0.238 | 0.008 | 0.124 | 0.173 |
| SMI (kg/m$^2$) | 0.276 | 0.002 | 0.075 | 0.414 |
| Trunk muscle mass per height squared (kg/m$^2$) | 0.315 | <0.001 | 0.098 | 0.281 |
| MEP (cmH$_2$O) | — | — | 0.550 | <0.001 |
| MIP (cmH$_2$O) | 0.550 | <0.001 | — | — |

Notes:
Controlling for sex and age.
BMI, body mass index; SMI, skeletal muscle mass index; MEP, maximal expiratory pressure; MIP, maximal inspiratory pressure.

while MIP does not. However, the association between these sarcopenia parameters and MEP/MIP is inconsistent across research populations (*Shin et al., 2017*; *Bahat et al., 2014*); thus, further research is needed. For the remaining respiratory function parameters, binomial logistic regression analysis did not show a significant association with sarcopenia in the overall and gender-specific analyses.

Our results showed that the cut-off value of MEP for predicting sarcopenia was 47.0 cm H$_2$0 for men and 40.9 cmH$_2$0 for women. Only one previous study with a Brazilian population calculated a similar cut-off value for MEP, which was 60.0 cmH$_2$0 for men and 50.0 cmH$_2$0 for women (*Ohara et al., 2018*). Our study demonstrates acceptable results because the respiratory muscle strength of Asian ethnic groups is low owing to their small body size (*Chen & Kuo, 1989*). FVC and PEFR, which are parameters of respiratory sarcopenia in previous studies, may be difficult to obtain for older adults even if they have no cognitive problems, because obtaining these parameters requires several repetitive breaths of varying intensity (*Nagano et al., 2021*; *Kera et al., 2019*). However, MEP only requires maximal expiratory effort from the maximal inspiratory position, which has the advantage of being easy to obtain.

This study has several limitations. Our participants were older adults requiring long-term care/support and may have been affected by disease conditions. However, it is difficult to determine whether sarcopenia and respiration are affected by primary (aging) or secondary (disease, etc.) factors. For this reason, we included them as control variables in the binomial logistic regression analysis. Since the multimorbidity of older Japanese adults was 62.8%, we could not eliminate disease-related factors (*Aoki et al., 2018*). The fact that the population had a high sarcopenia prevalence and a low AUC value, the generalizability of this study requires further survey among healthy individuals. Although MEP was found to predict whole-body sarcopenia in this study, these findings do not indicate that MEP is associated with future sarcopenic respiratory disability. Future longitudinal studies with a large sample size are needed to clarify the clinical significance of each parameter in respiratory sarcopenia.

## CONCLUSIONS

The most appropriate parameter for assessing respiratory sarcopenia may be MEP, which is an indicator of expiratory muscle strength, instead of FVC, MIP, or PEFR, as suggested in previous studies. It easy to instruct patients for MEP measurement. MEP is expected to be applied in clinical practice.

## ACKNOWLEDGEMENTS

Special thanks to all participants from the Nishinasuno General Home Care Center.

### Funding

This work was supported by the JSPS Grants-in-Aid for Scientific Research (grant numbers 20K07789 and 21K10581). The funders had no role in study design, data collection and analysis, decision to publish, or preparation of the manuscript.

### Grant Disclosures

The following grant information was disclosed by the authors:
JSPS Grants-in-Aid for Scientific Research: 20K07789, 21K10581.

### Competing Interests

The authors declare that they have no competing interests.

### Author Contributions

- Yohei Sawaya conceived and designed the experiments, performed the experiments, analyzed the data, prepared figures and/or tables, authored or reviewed drafts of the paper, and approved the final draft.
- Takahiro Shiba conceived and designed the experiments, performed the experiments, prepared figures and/or tables, authored or reviewed drafts of the paper, and approved the final draft.
- Masahiro Ishizaka conceived and designed the experiments, authored or reviewed drafts of the paper, and approved the final draft.
- Tamaki Hirose conceived and designed the experiments, performed the experiments, authored or reviewed drafts of the paper, and approved the final draft.
- Ryo Sato conceived and designed the experiments, authored or reviewed drafts of the paper, and approved the final draft.
- Akira Kubo conceived and designed the experiments, authored or reviewed drafts of the paper, and approved the final draft.
- Tomohiko Urano conceived and designed the experiments, authored or reviewed drafts of the paper, and approved the final draft.

## Human Ethics

The following information was supplied relating to ethical approvals (*i.e.*, approving body and any reference numbers):

The Ethical Review Committee of the International University of Health and Welfare approved this study (Approval No.: 17-Io-189-7).

## Data Availability

The raw measurements are available in the Supplemental Files.

## Supplemental Information

Supplemental information for this article can be found online at http://dx.doi.org/10.7717/peerj.12958#supplemental-information.

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
