# Peer review of "Sarcopenia is not associated with inspiratory muscle strength but with expiratory muscle strength among older adults requiring long-term care/support"

_PeerJ, doi:10.7717/peerj.12958_

## Round 0.1 · original submission · Minor Revisions

The two reviewers and I see much merit in this study and the resulting manuscript. Please look to take on board these reviewers' comments in the resubmission.

Reviewer 1 ·

Basic reporting

The paper is very well written, it is clear and concise. I have made a few minor recommendations, by line number below.

The introduction shows context and is well referenced with recent evidence. The structure of the article conforms to Peer standards. The figures are clear, well labelled, and relevant. Raw data was provided and appears to be well organised.

The term “daycare” may be better written as Day Care – the correct spelling appears to differ in different countries - this may be confirmed by the Academic Editor.

Experimental design

The research question is clearly defined, and it is clearly stated how this paper fills an identified knowledge gap.
The investigation performed a high technical standard, and the methods are described with sufficient detail to replicate the study.

Validity of the findings

Replication is encouraged.
The data appear to be robust and statistically sound.
Conclusions are well stated and linked to the research question.

Additional comments

Thank you for performing this fascinating study, you have addressed a genuine knowledge gap with innovation and expertise and discovered findings that are clinically relevant. I have made a few minor recommendations below:

Abstract
The conclusion ends rather abruptly, I recommend expanding on the final sentence to indicate how your finding/s may be clinically applied. You have used 234/500 words so there is plenty of room for more detail.
Introduction
Line 60 First sentence is a little disjointed. Please consider re-wording eg. “Respiratory sarcopenia has recently been defined as “whole- body…..”
Materials and Methods
The formatting appears to be different to the rest of the document.
Study participants
Please advise the rationale behind excluding those with respiratory conditions from your study.

Reviewer 2 ·

Basic reporting

Thank you for the opportunity to review this manuscript. Using a cross-sectional study, the authors investigated the association between respiratory parameters and sarcopenia. The topic is interesting, and the manuscript is well-written; however, there are some issues needed to be clarified.

Experimental design

1. The authors describe the baseline characteristics (Table 1) as well as AUC curve for sarcopenia by gender (Figure 2). It would be better to show the results by gender in tables 2 and 3.
2. The authors found that maximal expiratory pressure was the only parameter that was predictive of sarcopenia. They need to explain why maximal expiratory pressure but not other parameters were associated with sarcopenia.

Validity of the findings

1. The study was conducted among older adults who needed long-term care; thus, the findings may not be generalized to relative healthy individuals.

2. The authors aimed to identify cut-off for the obtained parameter for “the Japanese population”. I think they may need to tune down the statement about this given the small sample size (not representative) and cross-sectional design.

Additional comments

Minor concerns:
1. Given the high prevalence of sarcopenia (62.1%, 77 out of 124), I don’t think it is so meaningful to get AUCs of less than 0.7 (Figure 2).
2. Tables 2 and 3: why the authors indicate significant associations using asterisks if they already showed the exact p-values?
3. Did the authors test normal distribution for parameters such as trunk muscle mass per height squared and skeletal muscle mass index before performing correlation analysis?
4. I still don’t know how sarcopenia was defined. The authors may need to provide more details about it.

---

## Round 0.2 · Minor Revisions

Thanks for your attendance to the majority of the comments of the two reviewers. There are now just a small number of minor corrections required before this can be accepted for publication.

However, I forgot to mention this query from one of the reviewers in the previous review. Please also address this query regarding the ethics of your study:

I noted that The Ethical Approval Statement was provided in both English and Japanese and the participant consent form was only provided in Japanese (I am unable to interpret that). The title of the study on the Ethics Approval is “The Effect of the Difference in Muscle Mass, Respiratory Function, and Motor Function in the Elderly on the Food Form and Nursing Care Level.” This title is a little obscure and does not exactly line up (in my opinion) with the study here. The "Food Form" is not mentioned at all however, it could be that the paper submitted to Peer J is a smaller study from a larger overall body of works.

Reviewer 1 ·

Basic reporting

Thank you for your considered responses to both reviewers. I believe the article is now close to ready for publication and my only remaining comment is regarding the first sentence. I am unsure about the term “non-systematic review” and believe that detracts from the important opening paragraph. I recommend the following first sentence,

“Respiratory sarcopenia has been recently defined as “whole-body sarcopenia, and low respiratory muscle mass followed by low respiratory muscle strength and/or deteriorated respiratory function” (Nagano et al., 2021). (Lines 62-64)”

I have no further concerns or queries.

Experimental design

No further comments

Validity of the findings

No further comments

Additional comments

Thank you for conducting this interesting study.

Reviewer 2 ·

Basic reporting

No further comment

Experimental design

No more comment

Validity of the findings

No more comment.

Additional comments

The authors have well addressed my comments. I have only one concern. It is better to clarify how each component of sarcopenia was defined. For example, low muscle strength (hand grip strength) was defined by less than 30 kg in men and less than 20 kg in women.

---

## Round 0.3 · Minor Revisions

I think the authors for their attendance to the requests of the reviewers and myself. There is just one issue that still requires clarification. Within your definition of respiratory sarcopenia online 62 – 64 of the revised manuscript you describe it as including low respiratory muscle mass. However, the sarcopenia assessment and respiratory function and muscle strength sections of the revised manuscript (lines 116 – 139) do not describe any assessment for respiratory muscle mass. Please reconcile this difference in the definition of respiratory sarcopenia to your experimental approach.

---

## Round 0.4 · accepted · Accept

I thank the authors of this final amendment and am happy to recommend this paper for publication in PeerJ.